# Formation of an Amyloid-like Structure During In Vitro Interaction of Titin and Myosin-Binding Protein C

**DOI:** 10.3390/ijms26146910

**Published:** 2025-07-18

**Authors:** Tatiana A. Uryupina, Liya G. Bobyleva, Nikita V. Penkov, Maria A. Timchenko, Azat G. Gabdulkhakov, Anna V. Glyakina, Vadim V. Rogachevsky, Alexey K. Surin, Oxana V. Galzitskaya, Ivan M. Vikhlyantsev, Alexander G. Bobylev

**Affiliations:** 1Institute of Theoretical and Experimental Biophysics, Russian Academy of Sciences, 142290 Pushchino, Russiaogalzit@vega.protres.ru (O.V.G.);; 2Institute of Cell Biophysics, Federal Research Center Pushchino Scientific Center for Biological Research, Russian Academy of Sciences, 142290 Pushchino, Russia; 3Institute for Biological Instrumentation, Federal Research Center Pushchino Scientific Center for Biological Research, Russian Academy of Sciences, 142290 Pushchino, Russia; 4Institute of Protein Research, Russian Academy of Sciences,142290 Pushchino, Russiaalan@vega.protres.ru (A.K.S.); 5Institute of Mathematical Problems of Biology, Russian Academy of Sciences, Branch of the Keldysh Institute of Applied Mathematics, Russian Academy of Sciences, 142290 Pushchino, Russia; 6Branch of the Shemyakin–Ovchinnikov Institute of Bioorganic Chemistry, Russian Academy of Sciences, 142290 Pushchino, Russia; 7State Research Center for Applied Microbiology and Biotechnology, 142279 Obolensk, Russia; 8Gamaleya Research Centre of Epidemiology and Microbiology, 123098 Moscow, Russia; 9Scientific Center of Genetics and Life Sciences, Sirius University of Science and Technology, 354340 Sirius Federal Territory, Russia

**Keywords:** titin, myosin-binding protein C, amyloids, amyloid aggregation, protein aggregation, X-ray diffraction, atomic force microscopy, Fourier transform infrared spectroscopy

## Abstract

Protein association and aggregation are fundamental processes that play critical roles in a variety of biological phenomena from cell signaling to the development of incurable diseases, including amyloidoses. Understanding the basic biophysical principles governing protein aggregation processes is of crucial importance for developing treatment strategies for diseases associated with protein aggregation, including sarcopenia, as well as for the treatment of pathological processes associated with the disruption of functional protein complexes. This work, using a set of methods such as atomic force microscopy (AFM), transmission electron microscopy (TEM), Fourier transform infrared spectroscopy (FTIR), and X-ray diffraction, as well as bioinformatics analysis, investigated the structures of complexes formed by titin and myosin-binding protein C (MyBP-C). TEM revealed the formation of morphologically ordered aggregates in the form of beads during co-incubation of titin and MyBP-C under close-to-physiological conditions (175 mM KCl, pH 7.0). AFM showed the formation of a relatively homogeneous film with local areas of relief change. Fluorimetry with thioflavin T, as well as FTIR spectroscopy, revealed signs of an amyloid-like structure, including a signal in the cross-β region. X-ray diffraction showed the presence of a cross-β structure characteristic of amyloid aggregates. Such structural features were not observed in the control samples of the investigated proteins separately. In sarcomeres, these proteins are associated with each other, and this interaction plays a partial role in the formation of a strong sarcomeric cytoskeleton. We found that under physiological ionic-strength conditions titin and MyBP-C form complexes in which an amyloid-like structure is present. The possible functional significance of amyloid-like aggregation of these proteins in muscle cells in vivo is discussed.

## 1. Introduction

One of the factors determining the structural integrity and contractility of muscle tissue is the dynamic equilibrium between the synthesis, degradation, and aggregation of proteins in sarcomeres. Disruption of these processes can lead to decreased muscle elasticity, deterioration of contractility, and, as a consequence, age-related loss of mobility. Studying the interactions between titin and myosin-binding protein C (MyBP-C), which are important components of the sarcomere cytoskeleton, can provide new insights into the molecular mechanisms of maintaining the structural and functional integrity of sarcomeres in normal conditions and during aging and the development of pathological processes. In particular, the possible formation of amyloid-like structures (understood to be aggregates that have a cross-β structure (according to X-ray diffraction data) and the ability to bind ThT, but which do not form ordered fibrils) during the interaction of these proteins can play a role in stabilizing the sarcomere framework, and regulation of such interactions can be a key mechanism for preventing age-related degenerative changes. Considering that the efficiency of proteostasis systems decreases during aging [1] and that this contributes to the occurrence of chronic diseases caused by protein aggregation, the study of the mechanisms of myofibrillar protein aggregation is a high-priority scientific task.

Proteins are the most common biomacromolecules in the cells of living organisms. To perform their functions, proteins must remain in a stable folded (native) state. Association of proteins, leading to the formation of protein complexes, is necessary for the implementation of many biological processes important for the functioning of cells, organs, and the body as a whole. When proteins associate, interactions between them are specific and transient, and the formed protein complexes are subject to dissociation and reassociation [2,3].

One example of a polyfunctional multiprotein complex is the sarcomere, a highly ordered contractile unit of myofibrils of striated muscles of vertebrates (Figure 1). Although the protein composition of the sarcomere has been relatively well studied, there are currently very few data on the relative positions and interactions of proteins in different zones of the sarcomere, particularly in the A-zone, where titin, MyBP-C, and other proteins that form the basis of the sarcomere cytoskeleton interact with myosin filaments (Figure 1).

Interaction between titin and MyBP-C is known to play an important role in forming the sarcomeric cytoskeleton and providing the viscoelastic properties of muscle cells [4]. Titin and MyBP-C, which belong to the immunoglobulin superfamily, are multidomain proteins consisting mainly of immunoglobulin-like (Ig) [5] and fibronectin (type III)-like (FnIII) [6] domains. The titin molecule has the form of a filament more than 1 µm long, extending along half of the sarcomere from the Z-disk (N-terminus) to the M-line (C-terminus) (Figure 2). The MyBP-C molecule is a short filament 43 nm long, consisting of seven Ig-like domains and three FnIII-like domains. In the A-zone of the sarcomere, MyBP-C molecules interact with part of the titin molecule consisting of 11 domain super-repeats, the so-called C-zone (Figure 2). The exact positioning of MyBP-C on the titin molecule is described in [7]. According to the authors, three domains of MyBP-C (C8, C9, and C10) are associated with three domains of titin (Fn10, Fn11, and Ig1) (Figure 2).

Interaction between these proteins is an important factor in the integrity of the sarcomere, and mutations in their genes lead to severe pathologies [8,9,10,11,12,13]. Mutations in the titin gene in the A-zone of the sarcomere are most often pathogenic, since gene expression in this region is 100% [14,15,16,17]. Mutations in the *MYBPC3* gene encoding cardiac MyBP-C are associated with the development of various types of cardiomyopathies [12,13]. In this regard, the study of the structural features of the interaction of titin and MyBP-C is vital for understanding the fine mechanisms of protein–protein interactions in the sarcomere, which, in turn, can be important in choosing optimal approaches to the treatment of muscle diseases, primarily cardiomyopathies.

Our previous in vitro studies have shown that both titin and MyBP-C form amyloid aggregates in solutions with ionic strengths below physiological values [18,19,20,21,22]. An interesting feature of the aggregates is that, despite the presence of a cross-β structure, they are not highly ordered fibrils but have the appearance of amorphous aggregates. Moreover, amyloid aggregation of these proteins is not accompanied by changes in their secondary structure and is reversible: the aggregates disaggregate when placed in a solution with a high ionic strength [18,19,20,21,22]. We have suggested that only some regions of the titin or MyBP-C molecules are involved in the formation of the cross-β structure, while most of the protein sequence remains in the native conformation. The fact that there are no data in the literature on amyloid deposits of titin and MyBP-C in human or animal muscles suggests that amyloid aggregation of these proteins is possible in vivo and can be functionally significant. For instance, the functional role of amyloid aggregation of titin and MyBP-C in vivo can be to regulate the viscoelastic properties of muscle cells. Such regulation may be accomplished by forming a transient or permanent amyloid structure (cross-β structure) between molecules of interacting proteins.

In this work, using TEM, AFM, XRD, and FTIR techniques, as well as fluorimetric analysis using the dye thioflavin T, we conducted an in vitro study of joint aggregates of titin and MyBP-C formed under conditions close to physiological, with the aim of identifying structural features in them, including the presence of an amyloid structure.

## 2. Results

### 2.1. Electron Microscopy of Titin, MyBP-C, and Their Complexes

Figure 3A,B show micrographs of control samples of titin and MyBP-C in a solution containing 175 mM KCl and 10 mM imidazole, pH 7.0. Under these conditions, titin and MyBP-C form amorphous aggregates. Figure 3C shows micrographs of titin/MyBP-C complexes formed in a solution containing 175 mM KCl and 10 mM imidazole, pH 7.0, which exhibited significant morphological differences from the structures formed by titin (Figure 3A) and MyBP-C (Figure 3B) alone.

In particular, the micrographs of titin/MyBP-C complexes showed the presence of amorphous aggregates representing a structure resembling a bead network, which was uniformly distributed in a single layer over the observed surface (Figure 3C). Periodically, the field of view contained gaps in the form of dense formations—strands. The diameter of spherical aggregates (beads) was on average ~10–12 nm. Analysis of the ultrastructure of the obtained protein complexes suggests the interaction of molecules of titin and MyBP-C protein. In particular, we believe that MyBP-C molecules decorate titin molecules, forming on their surfaces structures resembling beads (Figure 3C, fragment 1). For a more detailed morphological characterization of the aggregates, a quantitative analysis of the images obtained by electron microscopy was performed. Three characteristic regions were identified (Appendix A), within which the diameter of individual globular structures was measured. The average values were as follows: region I, 36.9 ± 7.5 nm; region II, 29.9 ± 5.5 nm; region III, 28.4 ± 5.8 nm. This was especially visible in fragment 2 (Figure 3C), which showed a separately lying filament ~3 nm thick, presumably titin, decorated by another protein, most likely MyBP-C (shown in the figure with black arrows).

### 2.2. Atomic Force Microscopy

The titin/MyBP-C complexes were studied by atomic force microscopy. According to the AFM data, titin/MyBP-C complexes presented as a film distributed over the entire field of view with a height of up to 20 nm (Figure 4A,C). Single fragments of ~3–5 μm by ~30–40 nm can be observed near the edge of the film (Figure 4B). Quantitative analysis of AFM images (Figure 4A–C, Appendix A) showed that the surface formed by the titin/MyBP-C complexes is generally relatively flat, with local areas of increased and decreased relief. The average values of the aggregate height varied from 5.4 to 26.2 nm, with maximum values of up to 68.2 nm. The surface tilt angles did not exceed 0.1°, which also confirms the absence of a pronounced macrorelief. Thus, based on the set of parameters, it can be concluded that the aggregates form a relatively flat film with local areas of a surface uneven in height. This is consistent with the assumption of an amorphous or compact globular packing model.

### 2.3. X-Ray Diffraction Study of the Amyloid Properties of Titin/MyBP-C Complexes

It is known that amyloid proteins have a specific cross-β structure, which is characterized in the diffraction pattern obtained by X-ray diffraction by the presence of two reflections: ~10–12 Å and ~4.6–4.8 Å [23,24]. The ~10 Å reflection indicates the distance between β-sheets, while that of ~4.6 Å indicates the distance between the polypeptide chains in the cross-β structure.

X-ray diffraction analysis of the titin/MyBP-C complexes formed in a solution containing 175 mM KCl revealed two diffuse ring reflections at 4.7 and 10 Å (Figure 5A), characterizing the presence of a quaternary cross-β structure characteristic of amyloid fibrils. The blurring of the reflections can be explained by the absence of highly ordered structures (fibrils) in these complexes. No reflections were detected in amorphous aggregates of MyBP-C and titin formed in a solution containing 175 mM KCl and 10 mM imidazole, pH 7.0. It is known that for less oriented fibrils, both types of reflections attributed to the cross-β structure are blurred and appear in the diffraction pattern as rings [25]. In particular, a similar diffraction pattern was obtained for such a well-known amyloid protein as PmeL17 [26].

### 2.4. Binding of Titin/MyBP-C Complexes to Thioflavin T

The study of titin/MyBP-C complex binding to the ThT dye revealed an increase in fluorescence intensity compared to that in the presence of molecular titin and MyBP-C (Figure 5B).

Thus, the X-ray diffraction data together with the ThT binding data confirm that titin/MyBP-C complexes formed in a solution containing 175 mM KCl are of amyloid-like nature.

### 2.5. Secondary Structure of the Complexes Obtained by Fourier Transform Infrared Spectroscopy

FTIR was used to study changes in the secondary structure of proteins during the formation of the titin/MyBP-C complex.

Figure 5C shows the spectra obtained at 20 °C. The data show that the investigated samples of titin, MyBP-C, and their complexes had large percentages of disordered structure (70–80%, Appendix A). This can be observed from the characteristic features of the obtained spectra, the maxima of which have values of ~1640–1660 cm^−1^ (Figure 5C). The calculated values for the secondary structure components do not correspond to the literature data, since titin and MyBP-C are known to have a predominant β structure. Since not one but several possible peaks can be observed in the graph (Figure 5C), it was not possible to correctly calculate the values of the secondary structure elements, probably due to the complexity of the investigated objects. Therefore, to analyze the obtained data, we will take into account the characteristic features of the IR spectrum.

Molecular titin has an absorption spectrum of the amide I’ band with two broad peaks of equal intensity: 1628 cm^−1^ and 1640 cm^−1^ (Figure 5C, green curve). Molecular MyBP-C has a spectrum with a broad amide I’ band characterized by the absence of clear peaks (Figure 5C, blue curve). At the same time, low-intensity peaks were observed at 1628, 1645, and 1662 cm^−1^, which may correspond to minor β-structural components and loops or turns.

The titin/MyBP-C complex has a spectrum with a broad amide I’ band with a maximum at ~1640 cm^−1^ (Figure 5C, gray curve). An evident shoulder of the spectrum curve at 1628 cm^−1^ can also be noted. This spectral pattern indicates the presence of both native β-sheet conformations and intermolecular cross-β interactions, typical of amyloid-like assemblies. It is known that amyloid fibrils and native proteins with the β-sheet conformation have maxima within two characteristic, although partially overlapping, spectral regions. The spectral range characteristic of amyloid fibrils is between 1611 and 1630 cm^−1^, whereas native proteins with the β-sheet structure have amide I’ peaks located between 1630 and 1643 cm^−1^ [27,28]. The broader shape of the amide I’ band observed here is typical of non-fibrillar or partially ordered aggregates, consistent with other spectroscopic and diffraction data from this study. These FTIR results support the notion that titin/MyBP-C complexes consist of both native protein segments and cross-β structures, indicating the coexistence of intra- and intermolecular interactions during aggregation.

### 2.6. Prediction of Amyloidogenic Regions in Titin and MyBP-C

Amyloidogenic regions in titin and skeletal/cardiac MyBP-C were predicted using FoldAmyloid [29] and AGGRESCAN [30] programs. For titin (34,350 residues, containing 152 Ig and 132 FnIII domains) and MyBP-C (1141–1274 residues, containing 7 Ig and 3 FnIII domains), full-length sequences were analyzed. The amyloidogenicities of individual domains are shown in Figure 6.

For titin, AGGRESCAN predictions were obtained by dividing the sequence into 19 fragments due to software limitations on input length. FoldAmyloid processed the entire sequence at once. Amyloidogenic propensity was calculated as the ratio of amyloidogenic residues to total residues in the protein or domain. Average values were also calculated for individual domains, Ig domains, and FnIII domains (Table 1).

These findings indicate that both titin and MyBP-C contain multiple amyloidogenic “hot spots” within their domains, particularly in FnIII. Visual representation of domain-wise amyloidogenicity is shown in Figure 6.

## 3. Discussion

We have previously shown that both titin and MyBP-C are each capable of separately forming amyloid aggregates in vitro in solutions with ionic strengths below physiological values [18,19,20,21,22]. The distinctive features of the aggregation of these proteins are the following: a relatively high aggregation rate (large aggregates form within 10–40 min); no changes in the secondary structure of the proteins during aggregate formation; partial disaggregation. Based on the literature data showing that some domains of Ig and FnIII can undergo partial unfolding [31,32], and taking into account our own data showing the ease with which titin and MyBP-C form amyloid-like complexes, we assume that such interactions can occur without significant conformational rearrangements, potentially including partial domain unfolding as a structural trigger of the formation of intermolecular crosslinks.

Our FTIR data also support this assumption. When interpreting amide I bands, we relied on the criteria accepted in the literature: the range of 1630–1643 cm^−1^ is characteristic of native β structures, while that of 1611–1630 cm^−1^ is characteristic of intermolecular cross-β structures [27,28]. In the spectrum of the titin/MyBPC complex, the major maximum is located near 1640 cm^−1^, with a clearly defined shoulder at about 1628 cm^−1^, which indicates the simultaneous presence of native and amyloid-like β-components. We consciously used these ranges as an interpretative basis, without resorting to the deconvolution or the second derivative method, since references to the above-mentioned sources allowed us to reliably substantiate our conclusion.

Although the exact binding sites involved in the formation of amyloid-like complexes are unknown, the interaction between titin and MyBP-C appears biologically plausible given their well-established physiological co-localization within the sarcomere A-zone [7]. Our analysis shows that regions involved in binding, such as MyBP-C C8–C10 and titin Fn10, Fn11, and Ig1, contain amyloidogenic sites (Figure 6).

It is known that the propensity of proteins to aggregate is regulated not only by specific amino acid sequences, but also by the degree of amino acid sequence identity between adjacent domains in multidomain proteins. Previous studies on aggregation between different titin domains have shown that the probability of aggregation increases markedly when the sequence identity between adjacent domains exceeds 40% [33]. However, no similar calculations have been performed for domain comparisons between different proteins. In this work, we found that in the interaction region between titin and MyBP-C the amino acid sequence identity between their corresponding domains ranged from 20 to 30% (Appendix A). Interestingly, the presence of homologous domains in different proteins suggests their evolutionary relationship and points to a common path of structural and functional development. We suggest that this level of similarity can be sufficient to support a functional interaction while remaining below the threshold required for pathological aggregation.

It is well known that the key factors determining the intrinsic mechanical strength of amyloid fibrils are the intermolecular forces between β sheets and β strands [34]. A critical factor determining the mechanical properties of amyloid fibrils is the highly ordered hydrogen bonding network, in which interstrand hydrogen bonds act as a chemical “glue” enhancing the mechanical strength and structural stability of the fibrils [35,36]. Therefore, the putative functional role of the amyloid-like structures formed by titin and MyBP-C in vivo can be to maintain the integrity and ordered architecture of sarcomeres, thereby supporting the contractile capacity of muscle fibers. The presence of multiple such cross-β regions in the A-zone would likely endow myofibrils and muscle cells with the mechanical strength required to withstand repetitive mechanical loads.

In the context of our results, it is worth noting that protein–protein interactions have attracted considerable scientific interest in recent years. Considerable efforts have been directed towards the creation of a protein interactome, a comprehensive map of protein–protein interactions within a cell. Interactome mapping is considered fundamental for understanding the functioning of a living cell under both physiological and pathological conditions [37].

Understanding and predicting the mechanisms of protein interactions is of particular importance, as more than 80% of proteins function in molecular complexes [38]. More than 65 proteins are known to closely interact within the sarcomere [39]. Intra-sarcomeric protein interactions play a crucial role in maintaining sarcomeric structure and function, and their disruption can lead to pathological conditions such as myopathies and cardiomyopathies. The interaction between titin and MyBP-C is one such important interaction because it promotes sarcomere integrity, and mutations in either protein are associated with severe muscle pathologies [8,9,10,11,12,13,15]. In particular, mutations in the titin gene encoding the A-zone region are often pathogenic [14,15,16,17]. Similarly, mutations in the *MYBPC3* gene are among the most common causes of various forms of cardiomyopathy [8,9,13,40].

It has been suggested that titin and MyBP-C aggregates can accumulate intracellularly and inhibit the ubiquitin–proteasome system, thereby disrupting protein quality control mechanisms in cardiomyopathies [8,15,41,42,43,44,45]. However, the amyloid nature of such aggregates has not been confirmed. Given the demonstrated interaction of titin and MyBP-C in vivo, our results confirming amyloid-like co-aggregation of these proteins in vitro, and, more importantly, the lack of data on amyloid deposition of titin and MyBP-C in vivo, we propose that amyloid aggregation of these proteins can occur at sarcomeres and can be of functional rather than pathological significance [7,46]. At the same time, it should be noted that due to the strictly localized and limited number of titin and MyBP-C molecules involved in such interactions within the sarcomere, the existing histological methods, such as Congo red or thioflavin T staining, may not be sensitive enough to reliably detect such structures in vivo. Thus, the physiological significance of our in vitro data remains a hypothesis at this stage and requires further development of methodological approaches.

The formation and stabilization of amyloid-like titin and MyBP-C complexes in vivo may depend on physiological stress factors such as oxidative stress, impaired proteostasis, or age-related sarcomere remodeling. Aging muscle tissue is characterized by increased oxidative stress and decreased efficiency of the systems for removing damaged or aggregated proteins, including proteasomal degradation and autophagy, which increase the tendency for protein aggregates to accumulate [47,48]. In particular, accumulations of insoluble aggregates have been found in aging skeletal muscle, where their proportion in the insoluble fraction increases more than twofold [49]. Since amyloid-like structures are characterized by high stability due to dense β-structural packing and extensive intermolecular hydrogen bonds, these types of aggregates can be assumed to have a greater chance of accumulating in postmitotic tissue, such as skeletal muscle. With age, against the background of weakening of cellular control systems, the balance between reversible aggregation and the formation of stable structures can shift towards persistent, amyloid-like interactions.

The results of our study offer new insights into the role of amyloid-like structures in the sarcomere. In particular, we suggest that the formation of such structures can facilitate muscle adaptation to mechanical loading by increasing resistance to structural damage. This mechanism can play a key role in the age-related regulation of the structural and functional integrity of muscle tissue. Furthermore, given that physical activity promotes the synthesis and turnover of muscle proteins, future studies can focus on elucidating how mechanical loading affects the formation of amyloid-like structures between titin, MyBP-C, and potentially other sarcomeric proteins in vivo. Conversely, it cannot be ruled out that age-related weakening of the regulatory mechanisms controlling these processes can lead to the accumulation of non-functional/pathological aggregates, which in turn can worsen the elastic and contractile properties of muscle tissue. In this context, our results can help identify early disturbances in the interaction between the investigated proteins in the sarcomere and assess the risk of muscle dysfunction, especially in the context of aging or intensive physical activity.

## 4. Materials and Methods

### 4.1. Isolation of Titin and MyBP-C from Rabbit Skeletal Muscle

Titin and skeletal MyBP-C were isolated from rabbit skeletal muscles. All animal procedures were approved by the Commission on Biosafety and Bioethics (Institute of Theoretical and Experimental Biophysics, Russian Academy of Sciences, Permit No. 30, dated 10 September 2019) in accordance with Directive 2010/63/EU of the European Parliament. Rabbit surgeries were performed under anesthesia with Zoletil (Virbac Sante Animale, Carros, France) (4 mg/kg, i.m.); all efforts were made to minimize animal suffering.

Titin was isolated from rabbit hindlimb skeletal muscles according to [50] and purified by gel filtration on a Sepharose-CL2B column. Titin concentration was determined spectrophotometrically (Cary 100 UV–VIS spectrophotometer (Agilent Technologies, Santa Clara, CA, USA)) using an extinction coefficient (E280 mg/mL) of 1.37 [51].

Isolation of MyBP-C from rabbit skeletal muscles was carried out according to the method of [52] with minor modifications described in [20]. Protein concentration was determined spectrophotometrically (Cary 100 UV–VIS spectrophotometer (Agilent Technologies, Santa Clara, CA, USA) using an extinction coefficient (E280 mg/mL) of 1.09 [53].

The purity of the isolated titin and MyBP-C preparations was checked by SDS-PAGE, Western blotting, and HPLC-MS as described in [20,22] (Appendix A). For further studies, the protein preparations were lyophilized for subsequent dilution to the required concentration.

### 4.2. Electron Microscopy

Electron microscopic examination of the control preparations of titin and MyBP-C, as well as of titin/MyBP-C complexes, was performed in a solution containing 175 mM KCl and 10 mM imidazole, pH 7.0, in which none of the investigated proteins separately formed aggregates with an amyloid structure. The ratio of MyBP-C to titin was 1:2 at a total concentration of 0.1 mg/mL. The concentration of the control preparations of titin and MyBP-C was 0.1–0.2 mg/mL. Dialysis was performed for 24 h at 4 °C. A drop of the protein suspension was applied to copper grids covered with a collodion film reinforced with carbon. Then the grids were stained with a 2% aqueous solution of uranyl acetate. The samples were examined with a JEM-1200EX electron microscope (JEOL Ltd., Tokyo, Japan).

### 4.3. High-Performance Liquid Chromatography–Mass Spectrometry

Protein fractions were analyzed by tandem mass spectrometry. Proteins were treated with proteinase K (Promega, Madison, WI, USA) and trypsin (Sigma-Aldrich, St. Louis, MO, USA). Then the peptide mixture was separated by reversed-phase chromatography (Easy nLC1000; Thermo Fisher Scientific, Waltham, MA, USA) and analyzed by an OrbiTrap Elite ETD high-resolution mass spectrometer (Thermo Fisher Scientific, Waltham, MA, USA). The potential difference between the emitter and the inlet cone was 1.8 kV; the heated capillary temperature was 200 °C. Ion fragmentation was performed by collision activation in high-energy collisional dissociation and electron transfer dissociation. The masses of ions and ion fragments were recorded at resolutions of 60,000 and 15,000, respectively. The resulting fragmentation spectra were processed using the PEAKS Studio 7.5 software (Bioinformatics Solution Inc., Waterloo, ON, Canada).

### 4.4. Lyophilization of Proteins

To obtain a high concentration of MyBP-C and titin, the proteins were freeze-dried using a Free Zone 1 Liter Benchtop Freeze Dry System (Labconco, Kansas City, MO, USA). One percent trehalose was used as a stabilizing agent. The quality of proteins and any degradation after lyophilization were monitored in 7% SDS-PAGE [54].

### 4.5. Conditions for the Formation of Titin and MyBP-C Complexes

Lyophilized titin in a column buffer (0.6 M KCl, 30 mM KH_2_PO_4_, 1 mM DTT, 0.1 M NaN_3_; pH 7.0) was mixed with lyophilized MyBP-C in buffer (0.3 M KCl, 4.8 mM K_2_HPO_4_, 5.2 mM KH_2_PO_4_, 0.1 mM DTT, 0.1 mM NaN_3_; pH 7.0) at a ratio of 1:1. To form protein complexes, a protein mix was dissolved in deionized water and dialyzed in cellulose membrane tubing (size, 25 × 16 mm^2^) (Sigma-Aldrich, St. Louis, MO, USA) for 24 h at 4 °C against solutions containing 175 mM KCl and 10 mM imidazole, pH 7.0.

### 4.6. Atomic Force Microscopy

All samples were kept on mica for 5 min, washed twice with water for 30 s, and dried, and the structures of the obtained complexes were analyzed using the Integra-Vita system (NT-MDT, Moscow, Russia) using an NSG03 cantilever with an edge curvature radius of 10 nm and a resonance frequency of 47–150 kHz. Measurements were carried out in the semi-contact tapping mode. The obtained images were analyzed by the Nova program (Nova 1.0.26 NT-MDT, Moscow, Russia).

### 4.7. X-Ray Diffraction

Protein samples and control protein preparations, as well as their complexes, were concentrated using an Eppendorf 5301 Vacuum Concentrator (Eppendorf AG, Hamburg, Germany). The precipitate formed at the bottom of the tube was dissolved in an appropriate buffer solution to a concentration of 10 mg/mL. Then drops of the preparation were placed in the space (about 1.5 mm) between the ends of glass capillaries (about 1 mm in diameter) coated with wax. The X-ray diffraction data were collected on a Rigaku XtaLAB Synergy-S single-crystal X-ray diffractometer (Rigaku Corporation, Tokyo, Japan) equipped with a Hypix-6000C detector at the Collective Use Center “Structural and Functional Studies of Proteins and RNA” at the Institute of Protein Research, Russian Academy of Sciences (Pushchino, Russia), using the CrysAlisPro software, version 1.171.42.89a (Rigaku, Tokyo, Japan).

### 4.8. Thioflavin T Fluorimetric Assay

The ThT fluorescence intensity was studied by adding the dye to titin, MyBP-C, and titin/MyBP-C complexes. The weight ratio of ThT/protein was 1:5. The protein concentration was 0.1 mg/mL. Fluorescence was measured at λex = 440 nm and λem = 488 nm using a CaryEclipse spectrofluorimeter (Agilent Technologies, Santa Clara, CA, USA).

### 4.9. Fourier Transform Infrared Spectroscopy

FTIR spectra of solutions of titin, MyBP-C, and titin/MyBP-C complexes in a buffer containing 175 mM KCl and 10 mM imidazole, pH 7.0, and the spectra of the buffer solution itself were measured at 20 °C. Preparations of the titin/MyBP-C complexes were concentrated by diluting the lyophilized samples in the smallest amount (30–50 μL) of buffer solution. The protein concentration was 10–15 mg/mL. After dialysis of the control titin and MyBP-C preparations and co-dialysis of titin and MyBP-C, the samples were centrifuged. The resulting precipitates were diluted in 30–50 μl of the buffer solution.

The measurements were carried out on a Nicolet 6700 FTIR spectrometer manufactured by Thermo Scientific (Waltham, MA, USA), in transmission mode in a cuvette thermostatted at 20 °C with CaF_2_ windows and a sample thickness of 10 μm, using an MCT detector (liquid nitrogen cooling). Scanning was carried out in the wavenumber range from 1100 cm^−1^ to 4000 cm^−1^ with a resolution of 4 cm^−1^, averaging over 64 spectra. The optical path of the CaF_2_ cuvette was calculated for each measurement based on the optical density of the investigated sample at 3404 cm^−1^, using the absorption value of water at an optical path of 1 μm equal to 0.533 a.u., corrected for the concentration of protein in the sample [55]. The optical path of the cuvette was (10.1 ± 0.1) μm. The IR spectrum of the protein preparation was measured at least 10 times, and the spectrum of the buffer was measured the same number of times. The IR spectrum of the SP1 buffer was subtracted from each protein spectrum, taking into account the difference in the optical path values in the measurements. Each difference spectrum was corrected for the spectral contribution of water vapor and CO_2_ using the Omnic software, version 9.12.928 (Thermo Fisher Scientific, Waltham, MA, USA) followed by analysis in the range of the location of the Amide-1 bands at wavenumbers of 1720–1580 cm^−1^ for the content of secondary structure elements in the protein, following the principles described in [56]. The obtained estimates of the content of secondary structure elements in the protein were averaged.

### 4.10. Calculation of Amyloidogenic Regions

The amino acid sequences of human titin and MyBP-C of human skeletal and cardiac muscles were taken from the uniprot.org database (Q8WZ42, human titin; Q14324 and Q14896, MyBP-C of human skeletal and cardiac muscles). The length of the titin protein is 34,350 amino acid residues. It consists of 152 immunoglobulin and 132 fibronectin domains. The lengths of MyBP-Cs of human skeletal and cardiac muscles are 1141 and 1274 amino acid residues, respectively. They consist of 7 immunoglobulin and 3 fibronectin domains.

The amyloidogenicity of titin and MyBP-C of human skeletal and cardiac muscles was calculated using two programs: FoldAmyloid [29] and AGGRESCAN [30]. Amyloidogenicity was calculated as the number of amino acid residues included in the amyloidogenic regions divided by the total number of amino acid residues in the protein.

### 4.11. Amino Acid Sequence Identity

To reveal segments with large amino acid sequence identity that, according to available data, have an increased tendency for aggregation [57], we calculated the identity of the amino acid sequences between adjacent pairs of human titin domains (UniProt Q8WZ42), and cardiac MyBP-C (UniProt Q14896) were chosen for the calculations. Calculations carried out using the BLAST program (online tool available at https://blast.ncbi.nlm.nih.gov/ accessed on 15 July 2025) showed that the average identity of the amino acid sequences between neighboring FnIII domains did not exceed 33.7%, and that of those between neighboring Ig domains did not exceed 20.11%, which is a relatively low number (Appendix A).

## 5. Conclusions

In this work, we report for the first time the formation of amyloid-like structures in vitro at ionic strength values close to physiological in titin/MyBP-C complexes—proteins that interact in the sarcomere in vivo. The results obtained expand our understanding of the possible physiological role of these proteins and their interactions in sarcomeres. One of the possible functional significances of the aggregation of these proteins in vivo is the regulation of the viscoelastic properties of muscles during muscle contraction. Such regulation can be carried out by forming a transient or permanent amyloid structure (cross-β structure) between protein molecules. The data obtained open new vistas for studying the molecular mechanisms of muscle adaptation to physical loads. It cannot be ruled out that physical activity, being a natural way to maintain the necessary structural and functional integrity of the muscular apparatus, can influence the formation of amyloid-like structures in vivo, regulating the mechanical properties of sarcomeres. Further research should be aimed at identifying the relationship between physical activity, molecular changes in muscle, and active longevity processes.

## Figures and Tables

**Figure 1 ijms-26-06910-f001:**
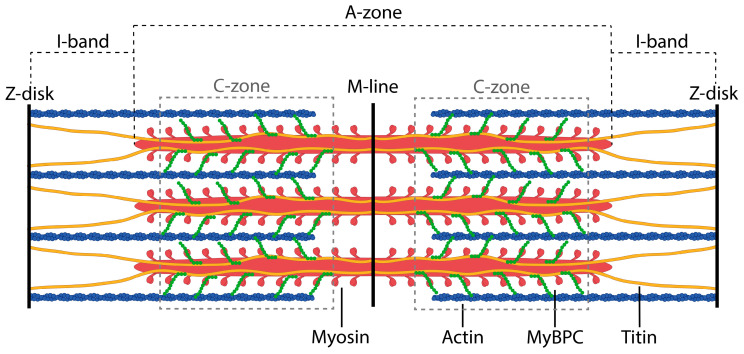
Schematic structure of a muscle sarcomere with thick and thin filaments. MyBP-C is localized in the C-zone and is highlighted in green; titin is shown in yellow.

**Figure 2 ijms-26-06910-f002:**
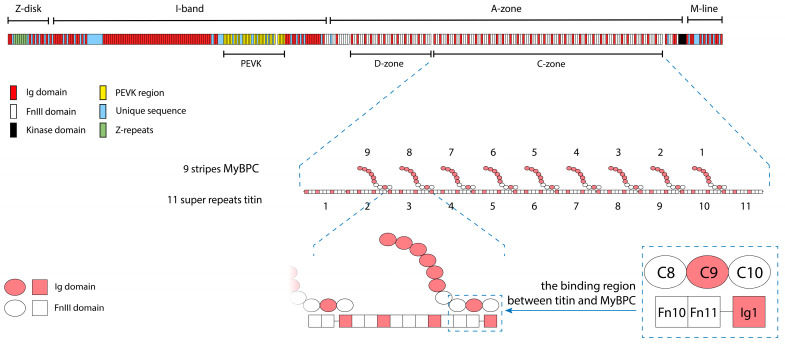
Localization of MyBP-C and titin molecules in the sarcomere. MyBP-C lies transversely to the axis of the thick myosin filament, forming 9 regularly located stripes with an interval of about 43 nm between the super-repeats of the C-zone of titin, starting from C2–C3 (the second and third super-repeats of the C-zone of titin) to C10–C11 (the 10th and 11th super-repeats of the C-zone of titin). However, between the last super-repeat of the D-zone and the first super-repeat of the C-zone of titin and between the super-repeats C1–C2, MyBP-C stripes are absent. According to the literature, three domains of MyBP-C (FnIII2, Ig7, and FnIII3) are associated with three domains of titin (FnIII10, FnIII11, and Ig1) at 11 sites of the titin molecule [4,7].

**Figure 3 ijms-26-06910-f003:**
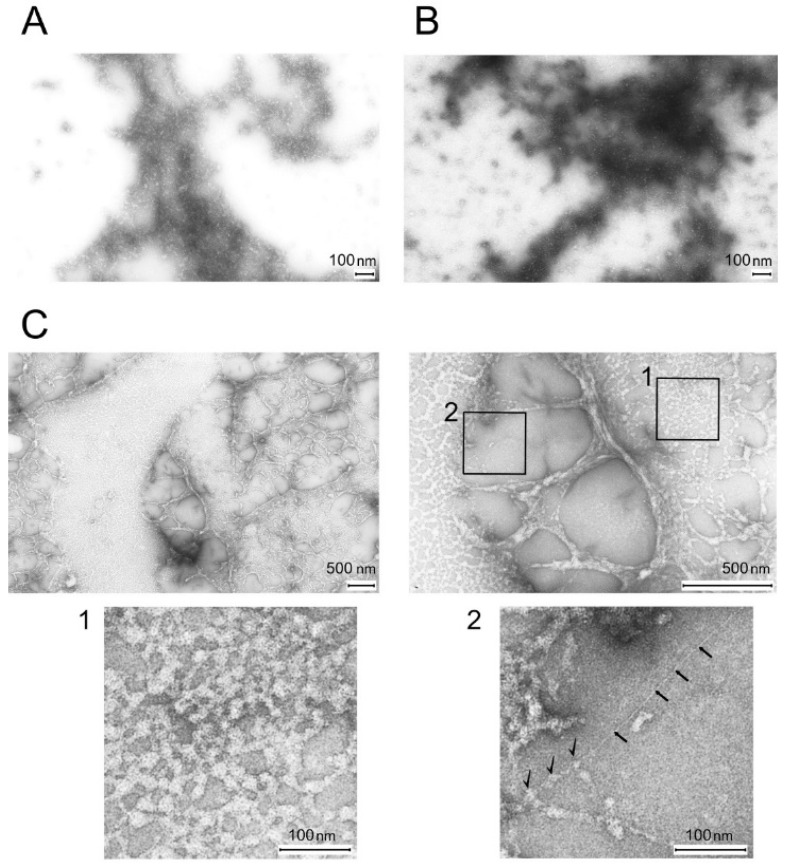
Electron microscopy of titin (**A**), MyBP-C (**B**), and titin/MyBP-C complexes (**C**) formed in a solution of 175 mM KCl and 10 mM imidazole, pH 7.0. Insets 1 and 2, enlarged views of informative regions. Inset 1, titin filaments presumably decorated with MyBP-C molecules. Inset 2, a single filament of presumably titin (marked with black arrows) emerging from the aggregate. Decoration elements, presumably MyBP-C molecules, on part of the filament (marked with black checkmarks). Quantitative analysis of electron microscopy images (Appendix A) showed that the average diameters of aggregates in regions I, II, and III were 36.9 ± 7.5 nm, 29.9 ± 5.5 nm, and 28.4 ± 5.8 nm, respectively, indicating a compact organization of the structures and a specific morphology of the detected aggregates. Negative staining with 2% aqueous uranyl acetate.

**Figure 4 ijms-26-06910-f004:**
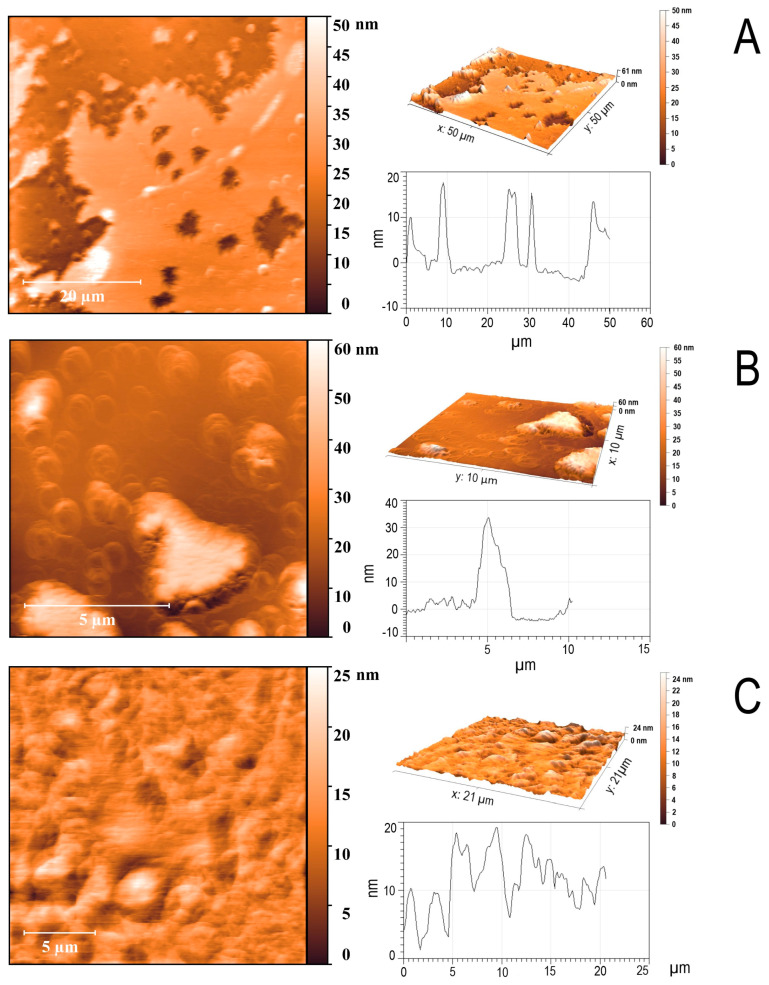
Atomic force microscopy of titin/MyBP-C complexes (**A**–**C**) formed in 175 mM KCl and 10 mM imidazole, pH 7.0. Dialysis was performed for 24 h at 4 °C. The ratio of MyBP-C to titin was 1:1. The final protein concentration was 0.12 mg/mL. Time of formation of aggregates and complexes at 4 °C, 24 h. Statistical analysis of the images (height, roughness, asymmetry, and tilt angles) is presented in the Appendix A. The obtained values indicate the presence of localized relief differences with an overall uniform surface, which corresponds to amorphous or compact aggregation.

**Figure 5 ijms-26-06910-f005:**
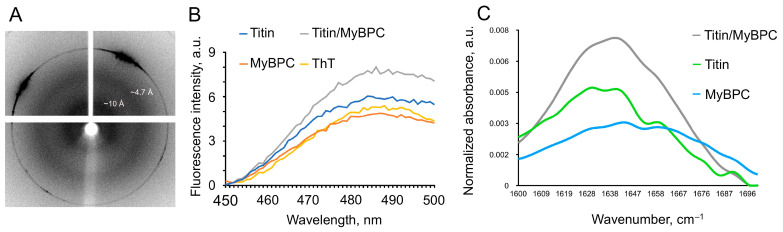
Confirmation of the amyloid-like structure in titin/MyBP-C complexes. (**A**) X-ray diffraction of titin/MyBP-C complexes after 24 h of co-incubation in 175 mM KCl and 10 mM imidazole, pH 7.0. Two diffuse ring reflections at 4.7 and 10 Å corresponding to the cross-β structure were detected. (**B**) Fluorescence intensity of ThT in the presence of titin (blue), MyBP-C (brown), and titin/MyBP-C complexes (gray) formed during 24 h in a solution containing 175 mM KCl and 10 mM imidazole, pH 7.0. The fluorescence intensity of ThT (yellow) was used as a control. (**C**) FTIR spectra of titin, MyBP-C, and titin/MyBP-C complexes at 20 °C. Protein concentration, 10–15 mg/mL. Green, non-aggregated skeletal muscle titin; blue, non-aggregated skeletal muscle MyBP-C; gray, titin/MyBP-C complexes formed in 175 mM KCl and 10 mM imidazole, pH 7.0.

**Figure 6 ijms-26-06910-f006:**
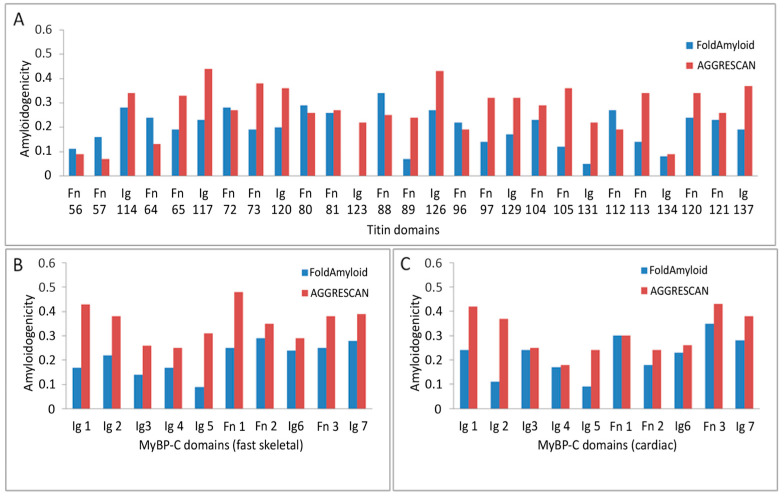
Predicted amyloidogenicity of titin and skeletal/cardiac MyBP-C. (**A**) Full-length titin (34,350 amino acid residues) composed of 152 immunoglobulin-like (Ig) and 132 fibronectin type III (FnIII) domains. (**B**) Skeletal MyBP-C (1141 residues) consisting of seven Ig-like and three FnIII-like domains. (**C**) Cardiac MyBP-C (1274 residues) with the same domain organization. Colored bars represent predicted amyloidogenic regions within individual domains, as determined by FoldAmyloid and AGGRESCAN algorithms. Highlighted domains correspond to known interaction sites between titin and MyBP-C.

**Table 1 ijms-26-06910-t001:** Amyloid propensities of titin and MyBP-C from human skeletal and cardiac muscles and average amyloid propensities of their individual domains.

**Titin**		
**Program**	**FoldAmyloid**	**AGGRESCAN ***
Whole protein	0.16	0.25
Domain average	0.19	0.27
Ig domain average	0.16	0.31
FnIII domain average	0.21	0.25
**MyBP-C (fast skeletal)**		
**Program**	**FoldAmyloid**	**AGGRESCAN**
Whole protein	0.18	0.29
Domain average	0.21	0.35
Ig domain average	0.19	0.33
FnIII domain average	0.26	0.40
**MyBP-C (cardiac)**		
**Program**	**FoldAmyloid**	**AGGRESCAN**
Whole protein	0.18	0.24
Domain average	0.21	0.31
Ig domain average	0.19	0.30
FnIII domain average	0.28	0.32

* AGGRESCAN estimates for titin obtained by averaging values across fragments.

## Data Availability

All data generated or analyzed in the course of this research (including files of additional information) were incorporated into the article and Appendix A.

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
