# Peer review of "Formation of an Amyloid-like Structure During In Vitro Interaction of Titin and Myosin-Binding Protein C"

_ijms, 2025, doi:10.3390/ijms26146910_

Round 1

Reviewer 1 Report

Comments and Suggestions for Authors

Review :  

The article is written well. The focus on  biophysical principles  controls the protein aggregation  and the diseases associated with protein aggregation  such as sarcopenia. Utilizing biophysical techniques such as  atomic force microscopy (AFM),  X-Ray diffraction,  Fourier Transform Infrared (FTIR), and electron microscopy in investigation the formation of amyloid like structure result from the interaction of titin and myosin-binding protein C (MyBP-C) strengthens the work and  offer new insights for future work of muscle dysfunction.

1)Abstract: There are abbreviations without definitions such as AFM, TEM, FTIR, X-RAY, and My BP-C. The abstract  lacks   brief details related to the results of  each biophysical techniques that are used in the study.  

2) Keyword: Some keywords have abbreviations, but others have full names. I suggest listing the names of the techniques.

3) Figure 4: B and C : The tiltle of  Y and x axis are not clear to read

4) In the discussion part, it is mentioned that ( previously shown that both titin and MyBP-C are able to form amyloid aggregates in vitro in solutions with ionic strengths below physiological values ). It would better to clarify if titin able to form amyloid and vitro and MyBP-C able to form amyloid separately  reference (18-  21).

5) In the discussion ( line 287- 291): It required reference 

6) in the line 234: Check (FoldAmyloid ). Is it Fold amyloid   

Author Response

We sincerely thank the Reviewers for their thoughtful and constructive comments. We carefully considered all suggestions and revised the manuscript accordingly. Each point is addressed in detail below.

Response to Reviewer 1

Reviewer: The article is written well. The focus on biophysical principles controls the protein aggregation and the diseases associated with protein aggregation such as sarcopenia. Utilizing biophysical techniques such as atomic force microscopy (AFM), X-Ray diffraction, Fourier Transform Infrared (FTIR), and electron microscopy in investigation the formation of amyloid like structure result from the interaction of titin and myosin-binding protein C (MyBP-C) strengthens the work and offer new insights for future work of muscle dysfunction.

Reviewer:

Abstract: There are abbreviations without definitions such as AFM, TEM, FTIR, X-RAY, and My BP-C. The abstract lacks brief details related to the results of each biophysical techniques that are used in the study.

Authors:

We expanded the list of abbreviations. The following was added to the Abstract (lines 33-40).

TEM revealed the formation of morphologically ordered aggregates in the form of beads during co-incubation of titin and MyBP-C under close-to-physiological conditions (175 mM KCl, pH 7.0). AFM showed the formation of a relatively homogeneous film with local areas of relief change. Fluorimetry with thioflavin T, as well as FTIR spectroscopy, revealed signs of an amyloid-like structure, including a signal in the cross-β region. X-ray diffraction showed the presence of a cross-β structure characteristic of amyloid aggregates. Such structural features were not observed in the control samples of the investigated proteins separately.

Reviewer:

Keywords: Some keywords have abbreviations, but others have full names. I suggest listing the names of the techniques.

Authors:

We have standardized the keywords to include full names of techniques without abbreviations, as suggested. The revised list now reads as follows:

Titin, myosin-binding protein C, amyloids, amyloid aggregation, protein aggregation, X-ray diffraction, atomic force microscopy, Fourier-transform infrared spectroscopy.

Reviewer:

Figure 4: B and C : The tiltle of Y and x axis are not clear to read

Authors:

The font size was increased, thank you!

Reviewer:

In the discussion part, it is mentioned that (previously shown that both titin and MyBP-C are able to form amyloid aggregates in vitro in solutions with ionic strengths below physiological values). It would better to clarify if titin able to form amyloid and vitro and MyBP-C able to form amyloid separately reference (18-21).

Authors:

The words “… are each capable of separately forming…” were added. (line 283)

Reviewer:

In the discussion (line 287- 291): It required reference

Authors:

Refs [7,54] were added, thank you!

Reviewer:

In the line 234: Check (FoldAmyloid ). Is it Fold amyloid

Authors:

The program’s name is FoldAmyloid, no space between the parts of the word.

Ref: Gurvich, A.V.; Baranov, P.V.; Stavropoulou, A.V.; Selivanova, O.M.; Goldman, A.; Gustafsson, C.; Zhouravleva, G.A.; Kisselev, L.L. FoldAmyloid: A Method of Prediction of Amyloidogenic Regions from Protein Sequence. Bioinformatics 2007, 23, 3405–3407.

https://doi.org/10.1093/bioinformatics/btm404

Reviewer 2 Report

Comments and Suggestions for Authors

This manuscript presents an intriguing study on the in vitro formation of amyloid-like complexes between two critical sarcomeric proteins: titin and MyBP-C. Using a comprehensive suite of biophysical techniques—including electron microscopy, atomic force microscopy (AFM), X-ray diffraction, FTIR spectroscopy, and thioflavin T (ThT) binding—the authors provide evidence of amyloid-like aggregation under near-physiological ionic conditions. Additionally, bioinformatics analyses predict amyloidogenic regions within titin and MyBP-C sequences, lending further support to their structural findings.

The work is well-conceived and addresses a novel angle in muscle biology by suggesting a possible physiological role for amyloid-like structures in sarcomere stability and function.

However, some critical aspects require further clarification and/or additional experimental support before the manuscript can be considered for publication.

Major Comments:

  1. The authors propose that amyloid-like interactions between titin and MyBP-C may occur in vivo and serve functional roles in muscle structure and mechanics. However, this claim is not supported by experimental data from tissue or animal models. The study would be significantly strengthened by including evidence from immunohistochemistry or Congo Red/ThT staining of muscle tissue to detect amyloid-like structures. Without such validation, the physiological relevance of the in vitro findings remains speculative.
  2. To support the biological significance of the findings, the authors should investigate whether similar titin–MyBP-C interactions or aggregates occur under physiological stress or aging. Proteomic analysis of muscle tissue under such conditions could reveal co-aggregation patterns or interaction signatures consistent with their in vitro observations, thereby reinforcing the functional relevance of the study.
  3. The manuscript currently presents microscopy and spectroscopic data in a largely qualitative manner. The authors should include quantitative analysis of aggregate size and morphology from electron microscopy or AFM images to strengthen the structural characterization. In addition, the FTIR spectra require more rigorous analysis—supporting the findings with second-derivative or deconvolution techniques would help distinguish native β-sheet elements from intermolecular cross-β structures characteristic of amyloid assemblies.
  4. The reported formation of titin/MyBP-C aggregates occurred at protein concentrations of 10–15 mg/mL. These levels appear significantly higher than what is typically present in muscle cells. The authors must justify the physiological relevance of these concentrations by providing quantitative estimates of titin and MyBP-C abundance in vivo
  5. It remains unclear whether the amyloid-like structures observed in this study result specifically from titin–MyBP-C interactions or are a consequence of general aggregation tendencies. The authors must include appropriate controls, such as incubating titin or MyBP-C with unrelated proteins, to determine whether co-aggregation is specific or non-specific in nature.

Minor Comments

  1. Please clarify early in the manuscript how “amyloid-like” is defined (e.g., based on cross-β structure, ThT binding, or morphology).
  2. Improve the clarity and labelling of figures, especially EM and AFM images, where identification of structures is sometimes ambiguous.

Author Response

We sincerely thank the Reviewers for their thoughtful and constructive comments. We carefully considered all suggestions and revised the manuscript accordingly. Each point is addressed in detail below.

Response to Reviewer 2

Reviewer:

This manuscript presents an intriguing study on the in vitro formation of amyloid-like complexes between two critical sarcomeric proteins: titin and MyBP-C. Using a comprehensive suite of biophysical techniques—including electron microscopy, atomic force microscopy (AFM), X-ray diffraction, FTIR spectroscopy, and thioflavin T (ThT) binding—the authors provide evidence of amyloid-like aggregation under near-physiological ionic conditions. Additionally, bioinformatics analyses predict amyloidogenic regions within titin and MyBP-C sequences, lending further support to their structural findings.

The work is well-conceived and addresses a novel angle in muscle biology by suggesting a possible physiological role for amyloid-like structures in sarcomere stability and function.However, some critical aspects require further clarification and/or additional experimental support before the manuscript can be considered for publication.

Reviewer:

Major Comments

Lack of in vivo validation of amyloid-like titin–MyBP-C complexes.

“The authors propose that amyloid-like interactions between titin and MyBP-C may occur in vivo... However, this claim is not supported by experimental data from tissue or animal models.”

Authors:

We fully agree with the reviewer that an in vivo confirmation would significantly strengthen the physiological significance of our observations. However, the presented study was initially designed as a biophysical in vitro study under conditions of ionic strength close to physiological values. We made respective clarifications in the revised Discussion, openly acknowledging this limitation.

We emphasize that the proposed physiological significance of amyloid-like interactions between titin and myosin-binding protein C remains a hypothesis at this stage, requiring further verification. Additional studies, such as Congo red staining of muscle tissue, ThT or immunohistochemistry, are indeed needed to assess the possibility of forming such structures in vivo.

At the same time, we would like to draw attention to an important methodological limitation: due to the strictly localized and relatively small number of titin and MyBP-C molecules in the sarcomere, visualization of such interactions using classical dyes may be technically unachievable at the current level of sensitivity. According to a recent cryoelectron microscopic analysis, there are only 6 titin molecules and 27 MyBP-C molecules per half of the myosin filament, strictly localized to the C-zone of the sarcomere (Tamborrini et al., 2023). This is a relatively small amount compared to other sarcomere proteins, and probably such an amount will not be enough to generate a signal detectable by standard amyloid-selective dyes designed for dense fibrillar aggregates.

Nevertheless, we share the reviewer’s opinion on the importance of such studies and we plan to conduct them in the future.

Ref: Tamborrini D, Wang Z, Wagner T, et al. Structure of the native myosin filament in the relaxed cardiac sarcomere. Nature 2023;623(7988):863-871; https://doi.org/10.1038/s41586-023-06690-5

Authors:

Added in Discussion, lines  354-360:

At the same time, it should be noted that due to the strictly localized and limited number of titin and MyBP-C molecules involved in such interactions within the sarcomere, the existing histological methods, such as Congo red or thioflavin T staining, may not be sensitive enough to reliably detect such structures in vivo. Thus, the physiological significance of our in vitro data remains a hypothesis at this stage and requires further development of methodological approaches.

Reviewer:

Lack of data on stress-induced or aging-related aggregation.

“To support biological relevance, the authors should investigate whether similar titin–MyBP-C interactions occur under physiological stress or aging.”

Authors:

We thank the reviewer for the valuable suggestion. Although such studies are beyond the scope of this paper, we have included a discussion of this prospect in updated Discussion. We assume that conditions such as oxidative stress, impaired proteostasis or age-related sarcomere remodeling may contribute to the formation or stabilization of amyloid-like protein structures in muscle cells. These assumptions open important avenues for future research, particularly in examining the effects of mechanical and metabolic stressors on titin and MyBP-C interactions in vivo.

Besides, we emphasized that truncated forms of titin and MyBP-C arising at pathogenic mutations may have an increased tendency to intracellular aggregation. “It has been suggested that titin and MyBP-C aggregates can accumulate intracellularly and inhibit the ubiquitin-proteasome system, thereby disrupting protein quality control mechanisms in cardiomyopathies [8,15,49–53]. However, the amyloid nature of such aggregates has not been confirmed.”

Authors:

Added in Discussion, lines 361-374:

The formation and stabilization of amyloid-like titin and MyBP-C complexes in vivo may depend on physiological stress factors such as oxidative stress, impaired proteostasis, or age-related sarcomere remodeling. Aging muscle tissue is characterized by increased oxidative stress and decreased efficiency of the systems for removing damaged or aggregated proteins, including proteasomal degradation and autophagy, which increases the tendency for protein aggregates to accumulate [55,56]. In particular, accumulations of insoluble aggregates have been found in aging skeletal muscle, where their proportion in the insoluble fraction increases more than twofold [57]. Since amyloid-like structures are characterized by high stability due to dense β-structural packing and extensive intermolecular hydrogen bonds, these types of aggregates can be assumed to have a greater chance of accumulating in postmitotic tissue, such as skeletal muscle. With age, against the background of weakening of cellular control systems, the balance between reversible aggregation and the formation of stable structures can shift towards persistent, amyloid-like interactions.

Refs:

  1. Kötter S, Krüger M. Protein Quality Control at the Sarcomere: Titin Protection and Turnover and Implications for Disease Development. Front Physiol. 2022 Jun 30;13:914296. doi: 10.3389/fphys.2022.
  2. Fernando R, Drescher C, Nowotny K, Grune T, Castro JP. Impaired proteostasis during skeletal muscle aging. Free Radic Biol Med. 2019 Feb 20;132:58-66. doi: 10.1016/j.freeradbiomed.2018.08.037.
  3. Ayyadevara S, Balasubramaniam M, Suri P, Mackintosh SG, Tackett AJ, Sullivan DH, Reis RJS, Dennis RA. Proteins that accumulate with age in human skeletal-muscle aggregates contribute to declines in muscle mass and function in Caenorhabditis elegans. Aging (Albany NY). 2016 Dec 15; 8:3486-3497 . https://doi.org/10.18632/aging.101141.

Reviewer:

Microscopy and FTIR data are largely qualitative.

“Include quantitative analysis of aggregates and rigorous FTIR spectral deconvolution.”

Authors:

We thank the Reviewer for a valuable comment. In response, we performed a quantitative analysis of electron microscopy images and included the results in a revised version of the manuscript and in the Supplementary file (Supplementary Figure S3). Three characteristic regions were identified within which the diameter of individual globular structures was measured. The average diameter values were: region 1, 36.9±7.5 nm; region 2, 29.9±5.5 nm; region 3, 28.4±5.8 nm. These data reflect the compact organization of the structures and the specific morphology of the detected aggregates.

Authors:

Added in Results, lines 152-157:

For a more detailed morphological characterization of the aggregates, a quantitative analysis of the images obtained by electron microscopy was performed. Three characteristic regions were identified (Supplementary Figure S3, Supplementary File S7), within which the diameter of individual globular structures was measured. The average values were: region I, 36.9±7.5 nm; region II, 29.9±5.5 nm; region III, 28.4±5.8 nm.  

Authors:

Added in the figure legend (Figure 3), lines 167-170:

Quantitative analysis of electron microscopy images (Supplementary Figure S3, Supplementary File S7) showed that the average diameters of aggregates in regions I, II and III were 36.9 ± 7.5 nm, 29.9 ± 5.5 nm and 28.4 ± 5.8 nm, respectively, indicating a compact organization of the structures and a specific morphology of the detected aggregates.

Authors:

(Аtomic force microscopy) Analysis of the AFM images (Fig. 4A–C) showed that the surface formed by the titin–C protein complexes is generally relatively uniform with moderate local height differences. In image 4A, the average height was 26.2 nm, with a maximum of 68.2 nm, but the average roughness (Sa) was relatively low at 7.1 nm. The surface tilt angles were also minimal (θ = 0.01°, φ = –0.58°), indicating that there were no pronounced slopes or inclined planes.

Similarly, in image 4B the average height is 5.4 nm with a maximum of 52.7 nm, but the roughness was only 7.5 nm. Here, local alternation of convex and concave areas is observed (Ssk = 1.95), but no global relief is revealed (θ = 0.09°). In image 4C (average height 16.1 nm), the surface was the most symmetrical in distribution (Ssk = –0.19), with minimal roughness (Sa = 3.2 nm) and zero slope (θ = 0.00°).

Thus, based on the set of parameters, it can be concluded that the aggregates form a relatively flat film with local areas of uneven surface height. This is consistent with the assumption of an amorphous or compact-globular packing model.

Authors:

Added in Results, lines 175-183:

Quantitative analysis of AFM images (Figure 4 (A–C), Supplementary Figure S5, Supplementary File S10) showed that the surface formed by the titin–MyBP-C complexes is generally relatively flat, with local areas of increased and decreased relief. The average values ​​of the aggregate height varied from 5.4 to 26.2 nm, with maximum values ​​of up to 68.2 nm. The surface tilt angles did not exceed 0.1°, which also confirms the absence of a pronounced macrorelief. Thus, based on the set of parameters, it can be concluded that the aggregates form a relatively flat film with local areas of a surface uneven in height. This is consistent with the assumption of an amorphous or compact-globular packing model.

Authors:

Added in the AFM figure legend (Figure 4), lines 188-191:

Statistical analysis of the images (height, roughness, asymmetry, tilt angles) is presented in the Supplementary Material (Supplementary Figure S5, Supplementary File S10). The obtained values ​​indicate the presence of localized relief differences with an overall uniform surface, which corresponds to amorphous or compact aggregation. 

Authors:

(FTIR deconvolution) We thank the reviewer for this constructive suggestion. We fully agree that second-derivative or deconvolution methods can improve the resolution of IR spectra, especially in the case of overlapping bands. However, in the present study we rely on the criteria outlined in authoritative and frequently cited papers (Barth, 2007 [37]; Sarroukh et al., 2013 [38]), which describe in detail that amide I band for amyloid structures is characteristic of the range of 1611–1630 cm⁻¹; for native β-sheets, 1630–1643 cm⁻¹.

In our spectra, the presence of a shoulder around 1628 cm⁻¹ is interpreted as a manifestation of cross-β interactions, and the main peak near 1640 cm⁻¹ reflects the retention of the native structure. We have clarified this interpretation in the revised manuscript and added direct references to these ranges in Discussion.

Authors:

Added in Discussion, lines 294-302:

Our FTIR data also support this assumption. When interpreting amide I bands, we relied on the criteria accepted in the literature: the range of 1630–1643 cm⁻¹ is characteristic of native β-structures; that of 1611–1630 cm⁻¹, of intermolecular cross-β structures [37, 38]. In the spectrum of the titin/MyBPC complex, the major maximum is located near 1640 cm⁻¹, with a clearly defined shoulder at about 1628 cm⁻¹, which indicates the simultaneous presence of native and amyloid-like β-components. We consciously used these ranges as an interpretative basis, without resorting to the deconvolution or the second derivative method, since references to the above-mentioned sources allow us to reliably substantiate our conclusion.

Reviewer:

Physiological relevance of protein concentrations used in vitro.

“The reported formation of titin/MyBP-C aggregates occurred at concentrations of 10–15 mg/mL, which seem high.”

Authors:

(FTIR concentration) We thank the reviewer for this comment. Indeed, the protein concentrations of 10-15 mg/mL used in the FTIR experiments are due to technical limitations of the method. Aqueous solutions have intense intrinsic absorption in the region of amide I band (~1650 cm⁻¹), and in order to register the protein spectrum against this background, a sufficiently high sample concentration is required.

As for the physiological relevance, it is important to note that the muscle cell sarcomere is a highly concentrated protein environment. According to the literature data, titin can account for up to 10% of the total myofibrillar proteins, and C-protein for about 2–4% [Gautel, 2011; Flashman et al., 2004]. Considering the strict localization of these proteins within the A-disk (C-zone) of the sarcomere, local concentrations near the filaments may be substantially higher than the average cytoplasmic values, but still lower than the concentrations used in our experiments.

Reviewer:

Specificity of titin–MyBP-C co-aggregation.

“Are the amyloid-like structures a result of specific interaction or non-specific aggregation?”

Authors:

Indeed, both titin and C-protein have a tendency to aggregate under certain conditions, as shown in our previous studies [18–22]. In this study, we performed control experiments in which titin and C-protein were incubated separately in a buffer with physiological ionic strength (175 mM KCl, pH 7.0). Under these conditions, each of the proteins formed only amorphous aggregates that did not have signs of an amyloid-like structure. This was confirmed by EM, AFM, ThT, FTIR and X-ray diffraction.

In contrast, co-incubation of titin and MyBP-C under the same conditions resulted in the formation of morphologically distinct bead-shaped aggregates with features of cross-β structure. These features were not observed in either individual samples or in earlier experiments at low ionic strength. Since titin and MyBP-C interact in vivo in the sarcomere A-zone, we believe that our in vitro data may indicate the biological significance and specificity of the interaction of these proteins in vivo. Although we did not test the interaction with unrelated proteins, the absence of amyloid-like features upon individual (separate) incubation of the investigated proteins indirectly confirms the specificity of the observed effect.

Reviewer:

Minor Comments

Define “amyloid-like” more clearly.

“Clarify early in the manuscript how ‘amyloid-like’ is defined.”

Authors:

In Introduction, we added a footnote on "amyloid-like": amyloid-like structures are defined as aggregates that have a cross-β structure (according to X-ray diffraction data), the ability to bind ThT, but do not form ordered fibrils. Thank you!

Reviewer:

Improve figure clarity (EM and AFM images).

“Improve labeling and clarity of EM/AFM figures.”

Authors:

Edits were made to increase the font size in the captions of Figure 4. The original EM image of titin and MyBP-C complexes was added to the Supplementary Figures file (Suppl. Figure S3).

Reviewer 3 Report

Comments and Suggestions for Authors

This study systematically investigated the amyloid-like structural characteristics of the titin and MyBP-C complex through a multidisciplinary approach, with the topic being innovative and having potential physiological significance. The experimental design is reasonable, and the overall data quality is reliable. I think the paper meets the publication standards of IJMS and I congratulate the authors for their interesting finding. I only have one minor comments:

The author mentioned C-zone in the Figure 1 legend, it should label it in Figure 1.

Author Response

We sincerely thank the Reviewers for their thoughtful and constructive comments. We carefully considered all suggestions and revised the manuscript accordingly. Each point is addressed in detail below.

Response to Reviewer 3

Reviewer:

This study systematically investigated the amyloid-like structural characteristics of the titin and MyBP-C complex through a multidisciplinary approach, with the topic being innovative and having potential physiological significance. The experimental design is reasonable, and the overall data quality is reliable. I think the paper meets the publication standards of IJMS and I congratulate the authors for their interesting finding. I only have one minor comments:

Reviewer:

The author mentioned C-zone in the Figure 1 legend, it should label it in Figure 1.

Authors:

The C-zone is now mentioned, thank you!

Round 2

Reviewer 2 Report

Comments and Suggestions for Authors

The authors have addressed all my previous comments thoroughly and made the necessary revisions to improve the manuscript. I believe the manuscript is now suitable for publication in its current form.